# DNA-Based Super-Resolution Microscopy: DNA-PAINT

**DOI:** 10.3390/genes9120621

**Published:** 2018-12-11

**Authors:** Daniel J. Nieves, Katharina Gaus, Matthew A. B. Baker

**Affiliations:** 1EMBL Australia Node in Single Molecule Science, School of Medical Sciences, University of New South Wales, Sydney, NSW 2052, Australia; d.nieves@unsw.edu.au (D.J.N.); k.gaus@unsw.edu.au (K.G.); 2ARC Center of Excellence in Advanced Molecular Imaging, University of New South Wales, Sydney, NSW 2052, Australia; 3School of Biotechnology and Biomolecular Science, University of New South Wales, Sydney, NSW 2052, Australia; 4CSIRO Synthetic Biology Future Science Platform, GPO Box 2583, Brisbane, QLD 4001, Australia

**Keywords:** DNA origami, DNA PAINT, DNA, super-resolution microscopy, SMLM, fluorescence microscopy

## Abstract

Super-resolution microscopies, such as single molecule localization microscopy (SMLM), allow the visualization of biomolecules at the nanoscale. The requirement to observe molecules multiple times during an acquisition has pushed the field to explore methods that allow the binding of a fluorophore to a target. This binding is then used to build an image via points accumulation for imaging nanoscale topography (PAINT), which relies on the stochastic binding of a fluorescent ligand instead of the stochastic photo-activation of a permanently bound fluorophore. Recently, systems that use DNA to achieve repeated, transient binding for PAINT imaging have become the cutting edge in SMLM. Here, we review the history of PAINT imaging, with a particular focus on the development of DNA-PAINT. We outline the different variations of DNA-PAINT and their applications for imaging of both DNA origamis and cellular proteins via SMLM. Finally, we reflect on the current challenges for DNA-PAINT imaging going forward.

## 1. Introduction

The discovery of fluorescent proteins, initially green fluorescent protein (GFP) from jellyfish [1], allowed the use of microscopy to genetically tag and visualize individual proteins. Consequently, a plethora of different colored fluorescent proteins were developed, which enabled labeling of multiple components within live cells for simultaneous observation. However, due to the diffraction limit of light (typically half the wavelength of the emission light), the observation of single molecules separated by less than this limit was a challenge. Recent developments in light and single molecule microscopy (culminating in the Nobel Prize for Chemistry in 2014) have enabled creative solutions to circumvent this diffraction limit and use light microscopy to resolve single molecules with nanometer accuracy [2,3,4], which became known as ‘super-resolution’ microscopy.

The technological basis of one implementation of super-resolution microscopy is the stochastic sampling of subsets of excited molecules within an image. This allows the location of each individual fluorophore to be measured with precision, so that each fluorophore is precisely localized, known as single molecule localization microscopy (SMLM). The precision of localization depends only on the number of photons (*N*) we can observe and improves as 1/√*N*. The two more commonly used techniques for SMLM are photoactivatable and photoconvertible localization microscopy (PALM) and stochastic optical reconstruction microscopy (STORM). Photoactivatable and photoconvertible localization microscopy requires expression of the protein of interest in fusion with a photoactivatable fluorescent protein. The blinking mechanism involves using light to stochastically switch the dyes from a ‘dark’ to a ‘fluorescent’ state, e.g., photoactivatable GFP has a low emission when excited at 488 nm and increases ~100-fold when activated with 405 nm light [5]. Alternatively, fluorophores can be converted from one color to another, e.g., monomeric Eos2 (mEos2) emits at 516 nm and then 581 nm when photoconverted with UV light (~390 nm) [6]. Stochastic optical reconstruction microscopy is another approach to SMLM, which takes advantage of the photophysical properties of organic dyes to induce a blinking effect [7]. Unlike PALM where the fluorescent proteins are inherently designed to switch between various states, STORM requires a combination of reagents in specific buffers to manipulate the photophysical pathways of fluorescent organic dyes [8]. The use of organic dyes means that STORM produces images with higher localization precision because the quantum yield for organic dyes is higher in comparison with fluorescent proteins [9]. This improves image resolution. Both PALM and STORM have advanced the understanding of biological complexes; however, both only allow a limited number of observations for each labeled molecule, due to the constraints imposed by the photon budget of a fluorophore, i.e., the number of photons a fluorophore can emit before bleaching. This can lead to under sampling of the true underlying biology. Techniques that allow multiple repeated observations of molecules, unrestricted by the photon budget, such as point accumulation for imaging in nanoscale topography (PAINT) [10] are thus attractive for SMLM.

## 2. PAINT

An alternative method for achieving blinking of molecules is to have permanent or transient binding of a fluorophore to a target of interest. This approach often involves a pool of fast diffusing fluorescent ligands in the medium that are imaged only when bound to their target on the cell, since this target is either immobile or slowly diffusing on the membrane. The first iteration of this was by Sharonov and Hochstrasser [10], who used intermittent collisions between Nile Red and large unilamellar vesicles (LUVs) as the basis of their imaging (Figure 1). The intermittent collisions, localization, and subsequent photobleaching enabled them to assemble a super-resolution image. They introduced the terminology of ‘point accumulation for imaging nanoscale topography’ (PAINT). This work did not rely on specific or tunable interactions between the imager (in this case Nile Red) and the target (the LUVs). In this case the interaction was the hydrophobic interaction of dye with the inside of the membrane bilayer, thus, it was irreversible and bleaching of the bound molecules was required. Subsequent work generalized this approach to allow dynamic imaging of other biomolecules continuously and stochastically with fluorescent ligands in solution [11]. For example, the interactions between Ni-tris-NTA on the imager and a target protein with a His-tag were a suitable combination for PAINT, and this was termed as universal PAINT (uPAINT). While uPAINT generalized this approach to specific biomolecules: using defined ligand/receptor pair: antibody-epitope [11], receptor ligand [12], and Nanobody/GFP [13]. This enables one to obtain super-resolved images in fixed cells and diffusion maps of membrane proteins in living cells [11,14]. The latest incarnation of these PAINT approaches uses custom fluorophores engineered at Janelia Farm that bind transiently to specific targets [15]. Spahn et al. recently introduced a range of known and novel rhodamine-based probes to facilitate PAINT imaging and combined PAINT imaging with photoactivatable fluorescent proteins to visualize RNA polymerase distributions relative to nucleoid [15,16].

DNA is a natural candidate for developing a method for PAINT imaging as it offers a sequence specific, precisely tunable interaction between two oligomers of single stranded DNA. This can be used as the basis for a controlled, transient interaction, and optimized for single molecule imaging and SMLM. This technique was first demonstrated by Jungmann et al. in 2010 [17].

## 3. DNA as a Programmable Polymer

The DNA molecule has several features that make it particularly appealing for strategic self- assembly, particularly due to the well understood, strong, and specific interaction with a complementary strand, based upon sequence identity. Additionally, with a diameter of ~2 nm, a helical pitch of ~3.4–3.6 nm and a persistence length [18] of approximately 50 nm, it is highly suited for construction on the nanoscale.

These factors, coupled with the ease of synthesis of nucleic acids [19] and the ability to conjugate functional groups and other covalent modifications [20] have placed DNA self-assembly at the forefront of synthetic self-assembling nanotechnology, since its inception less than 30 years ago by Ned Seeman who first realized that DNA lattices could be engineered with a structural purpose [21] (Figure 2). 

Pioneering work by Paul Rothemund enabled larger structures to be made with DNA whilst retaining nanometer precision [23]. The principle of the DNA origami technique is to take a long scaffold strand, commonly a viral genome, e.g., M13mp18 with 7249 nt, and fold it over multiple times using staple strands of around 30 nt in length (Figure 1). These structures normally require approximately 200 staple strands, which specifically hold together different sections of the long strand, folding it into a structure. From initial work on DNA systems and construction of 3D nano-structures such as cubes [25] and tetrahedra [26], the applications of DNA nanotechnology are now diverse, including: DNA tweezers [27], DNA-based molecular walkers [28], nanoparticle organization [29], enzyme-free logic circuits [30], controlling biosynthesis pathways [31], and targeted drug delivery to cells [32].

Additional properties of DNA that make it suitable to design and fabricate self-assembling nanostructures, the specificity and predictability of Watson-Crick base pairing between individual single strands of DNA makes it highly suited for an application such as DNA-PAINT. The logarithmic dependence of affinity on sequence length has been well characterized due to the importance of single stranded overhangs, known as toe-holds, in driving DNA functionality [33]. Thus, an imager-docking sequence interaction based on the base pairing of 9–10 nucleotides, typical for DNA-PAINT applications, has a *k*_on_ of 10^6^ M^−1^ s^−1^.

One can sequentially label a limited number of sites with fluorescent-oligonucleotides (length > 10 bp) resulting in a strong binding interaction with a long hybridization time. This is the basis of a technique known as fluorescent in-situ hybridisation (FISH), which was initially used to karyotype human chromosomes using multiple colors of fluorophores [34], and now is used for a range of DNA imaging applications [35]. More recently FISH has been used in combination with 3D structured illumination microscopy (3D-SIM) [36], an alternative implementation of super-resolution microscopy, whereby the sample is illuminated with multiple interfering light beams and the interference in the emitted fluorescence, similar to Moire fringes, which can be used to extract higher spatial information. Here, single stranded DNA handles from between 30–1000 nt in length were used as the target fluorescently labeled complementary DNA strands. DNA-PAINT, in contrast, typically uses imager-docking strand pairings that are 9 or 10 nt in length, and remain bound for <2 s [17].

## 4. Sub-Diffraction Imaging of DNA Origami via DNA-PAINT

Jungmann et al. provided the first integration of PAINT imaging with DNA origami and showcased the use of fluorescent DNA oligomers to provide precise, transient interactions between docker and imager [17]. Jungmann et al. used long rectangular origamis (approx. 20 nm × 260 nm) with 3 docking strand sites spaced approximately 130 nm apart to show that the use of complementary dye-labeled imager DNA was able to resolve these separate sites (Figure 1j) [17]. This work demonstrated that the repeatable binding and unbinding from the target strands could be exploited to enable sub-diffraction imaging, termed DNA-PAINT (Figure 3). Since then, origamis have also become a useful metrology platform for demonstrating other variations of DNA-PAINT imaging, nominally quantitative PAINT (qPAINT) [37], Förster resonance energy transfer PAINT (FRET-PAINT) [38], and Exchange PAINT [39,40] reviewed here in later sections.

DNA-PAINT is a particularly well-suited platform for the characterization of DNA origami structures by fluorescence-based microscopy, both as a means to test instrument performance and origami folding/strand incorporation efficiency. DNA-PAINT imaging was implemented to observe polyhedral-like origamis, including tetrahedrons, triangular prisms, cubes, pentagonal prisms, and hexagonal prisms, all with edge lengths of 100 nm and with target strands at their vertices (Figure 3f,g) [41]. Here, Iinuma et al., were able to clearly resolve the vertices, and thus the quality of the formed structure, via a 3D implementation of DNA-PAINT, thus allowing characterization of the near-solution shape of DNA origami structures [41]. Similarly, origami structures with target sites separated by less than 10 nm provided a useful tool for microscopists to push the limit of precision and to resolve adjacent targets on their systems. DNA origami had already been used as super-resolution rulers by Steinhauer et al., but using STORM imaging, with sites spaced approximately 90 nm apart [42]. This approach was limited due to STORM requiring a fixed dye label, thus, the number of localizations that could be observed before bleaching was limited. However, Raab et al. demonstrated a dramatic improvement on this by using DNA-PAINT to resolve targets spaced 6 nm apart [43]. This demonstrated the significant advantage of DNA-PAINT—that unlimited observations, and thus localizations, can be recorded due to repeated binding of a dye labeled imager. This was further explored by Dai et al., using origami tiles with densely packed targets separated by 5 nm (Figure 3h) [44]. DNA-PAINT imaging, in conjunction with filtering to retain the localizations of highest quality (e.g., highest precision), and summation of images across multiple identical origami structures allowed the authors to resolve the closely spaced sites (Figure 3h) [44].

DNA-PAINT imaging of origami structures has also revealed that the interaction of docking strand and imager could be altered depending on the location of the docking strand. Kinetic rate constants for the long rectangular origamis were comparable to those obtained from single immobilized strands at a surface; however, for the regular rectangular origamis the *k*_on_ was reduced along with a 30% increase in the *k*_off_ for docking strands located at the center of the origami (Figure 1j) [17]. This suggested that the conformation of the origami when immobilized could have an effect on single strand binding kinetics. Quantitative counting of stoichiometry using single strand kinetics, as discussed in the next section (qPAINT), is robust to these variations, at least as far as detecting integer changes in stoichiometry with docking strands as close as 6 nm. Furthermore, DNA-PAINT imaging has become a useful tool for probing staple strand incorporation for DNA origamis. During the development of DNA-PAINT, Jungmann et al. observed that their rectangular tile origami that had been designed to have 12 docking sites was often imaged containing 11 sites or fewer [37]. It was noted that this could be a result of inefficient incorporation of target strands within the origami structure. Building upon this work, Strauss et al. probed the incorporation efficiency of target strands at every position on a rectangular origami (Figure 3i) [45]. It was observed that the incorporation efficiency of targets was strongly influenced by its position on the origami. Minimum strand incorporation was observed at the periphery of the origami, with some strands incorporated at a rate as low as 40–50%. In contrast, it was observed that the sites with the highest incorporation lay at the middle of the tile, with a maximum of 95% [45]. DNA-PAINT has provided here a unique insight into origami design, informing where strands that are crucial to origami function should be placed.

## 5. Counting Molecules/DNA-Docking Strands—Quantitative DNA-PAINT (qPAINT)

The simplest method to count the stoichiometry of the number of docking sites is to individually resolve them in a super-resolution image and count the number of sites. However, this is only suited to well-spaced docking strands, and also can be misleading when the true number of sites is not known [45]. DNA-PAINT naturally offers an alternate method to measure stoichiometry via the information gleaned from the frequency with which the imager binds to the docking strand during DNA-PAINT imaging [17,37]. The binding and dissociation of a single DNA imager strand to a docking strand follows a simple second-order association rate constant *k*_on_ and a first-order dissociation rate constant *k*_off_, respectively. These kinetic constants determine the duration of the fluorescence ‘on-’ (imager bound) and ‘off-times’ (imager unbound). The ‘on-time’ is equal to *k*_off_^−1^, whereas the ‘off-time’ is equal to (*k*_on_ × *c*_i_)^−1^ where *c*_i_ is the concentration of imager strand [37]. Therefore, the frequency of ‘blinking’ of a given structure is linearly proportional to the underlying number of docking strands (Figure 4). In order to extract the number of sites within a given region or complex of interest time traces are acquired over hours (given the ‘off-time’ can be on the order of tens to hundreds of seconds for a single site [46]) and it is the time between ‘blinks’ that is recorded. These ‘off-times’ are then used to build a cumulative distribution function, which can be fitted with a single exponential to extract the mean ‘off-time’. This ‘off-time’ can then be converted into a stoichiometry of docking strands by comparison to the ‘off-time’ for a known calibration, such as a DNA-origami structure with a known number of sites [37].

This quantitative method of PAINT, qPAINT, provides an orthogonal method for extracting and comparing quantitative information from SMLM data. Recently, it was applied to investigate the distribution of signaling ryanodine receptors (RyRs) within the membrane of cardiac myocytes [47]. Jayasinghe et al. were able to resolve single RyRs within the plasma membrane and reveal that they are distributed within irregular clusters. It was also observed that a RyR inhibitory protein, junctophillin-1, co-clustered with RyRs, and that the stoichiometry between the two proteins was highly heterogeneous (junctophillin-1:RyR ratios varied from 2:1 to 2:7), thus, suggesting an additional layer of complexity in the regulation of RyR [47]. This application shows the potential for qPAINT to reveal stoichiometric relationships between interacting proteins within cells.

## 6. Exchange PAINT

The ability to tune and vary the DNA-PAINT docking sequence allows the observation of multiple, and mutually exclusive docking strands, at high resolution within the same sample and using the same fluorophore. This method, known as Exchange PAINT, relies on the high number of orthogonal sequences that can be constructed using nine nucleotides such that multiple sites can be labeled with unique and non-overlapping docking sequences [39]. Thus, many different docking strands can be imaged using the same fluorescence channel by washing in and out different complementary imager strands (Figure 4d). This was first demonstrated by imaging each label in DNA-origamis labeled with 10 unique docking strands sequentially [39]. This multiplexing capacity is limited only by the large number of possible orthogonal 10 nt sequences and is achieved with relative ease in comparison with fluorescence methods which require targets to be spectrally separated.

## 7. FRET-PAINT

DNA-PAINT imaging has been modified to exploit fluorescence resonance energy transfer (FRET), to overcome some of the disadvantages over the conventional implementation [49]. Even though DNA-PAINT imaging is governed by highly specific interactions between DNA strands, the addition of, often hydrophobic, dyes can cause a moderate level of non-specific background within the images. Similarly, the high excess of dye labeled imager in solution (~500 pM–10 nM), which has a fluorescence emission while diffusing rapidly, can contribute to an overall higher background in the images and in turn decrease the precision of detection of bound imager strands. Recently, Auer et al., implemented two FRET approaches in an attempt to reduce background contributions from unbound imager strands [38]. The first has an acceptor dye on the docker strand, and the donor free in solution on the imager strand. In this configuration emission in the acceptor channel is only seen when the donor imager strand is bound. The second employs a longer ~25 nt docking strand that can allow the anti-parallel binding of two imaging strands the imager mix (a donor and an acceptor; Figure 4b), termed “dynamic” FRET-PAINT. Thus, a signal in this case is only generated in the acceptor channel when both are bound to the docking strand. This enables essentially ‘background-free’ imaging due to the requirement for both the acceptor and donor to be bound at the docking in order for any emission to occur [38,49]. More recently, this approach was further refined by measuring the FRET efficiency of acceptor donor strand interactions allowing super-resolution FRET imaging. It was also observed that FRET efficiency could be used to differentiate docking sequences with different binding geometries, i.e., different distances between donor and acceptor fluorophores [50].

## 8. Cellular Imaging via DNA-PAINT

DNA-PAINT-based imaging approaches present a variety of options for imaging single molecules, and this has been readily exploited for cellular imaging. One of the key challenges when adapting DNA-PAINT for imaging proteins or structures in biological samples is the need to label the molecules of interest with a DNA docking strand. Conventional approaches have succeeded here, for example, the use of antibodies conjugated via heterobifunctional chemistry to a DNA docking sequence. This is a popular avenue for imaging of multiple species within cells in combination with Exchange PAINT. This approach has been demonstrated for imaging of up to eight separate proteins/targets in vitro to image a variety of proteins in many different cells and samples [51]. For example, as a proof of principal, Wang et al., were able to image six separate targets from different subsections in a mouse retina tissue section. Thus, this approach can give a great amount of information on structures and features of specimens at the micron scale.

Similarly, this approach has been applied for single molecule imaging of structures at the nanoscale. The potential for imaging cellular structures using DNA-PAINT was first demonstrated using multiplexed Exchange-PAINT imaging [39]. Here, the authors were able to image the mitochondria and microtubules in fixed cells by labeling each structure with antibodies conjugated to non-identical docking sequences. In the same work this was extended to allow 3D imaging of such structures by SMLM, thereby being able to resolve the z-position of the structures with high precision [39]. This approach was adapted to a spinning disk confocal microscope, which allowed imaging of thick samples [52]. Scheuder et al. demonstrated using again mitochondrial and microtubules, with the addition of HSP-60 proteins, that whole cells could be imaged with this method. It was also demonstrated that FISH-type experiments could be performed, similar to previous studies [53,54], allowing the super-resolution of DNA and mRNA in fixed cells [52]. Here, 3D STORM imaging was achieved by oblique illumination of the sample, and the dye labeled oligos were localized using a 3D localization algorithm [55]. Recently, the reconstruction of thick 3D samples has been realized using SMLM and electron multiplying charge coupled device (EMCCD)-based detection, as opposed to confocal approaches. The imaging was achieved by scanning of line-focused laser light across the sample by a galvanometric mirror, with the emitted light reflected by the same mirror and projected through a confocal slit. This is coupled synchronously with a second galvanometric mirror on the detection path that focuses the light from the slit onto the EMCCD. Here, microtubules and mitochondria were again the structures used to validate the reconstruction method and were imaged up to 6 µm into the sample [56]. DNA-PAINT imaging has also provided a chance to observe cell surface receptors such as ryanodine receptors (RyRs) [47], epidermal growth factor receptors (EGFRs), insulin-like growth factors (IGFs) and the protein family ErbB (Figure 5) [48]. In the case of EGFRs, Exchange-PAINT was used to image the EGFR along with ErbB2, ErbB3, IGF-1R and Met [48]. This revealed that these receptors were non-homogenously distributed on the plasma membrane and were intermixed (Figure 5a). It was observed that stimulation with EGF did not induce any changes in the distribution of EGFR, but did lead to agglomeration of EGFR with both ErbB3 receptors and Met [48].

Due to the interest in observing single receptor proteins, there has also been a drive to develop new strategies to label single proteins with a single docking strand. These approaches seek to reduce the linkage error in single molecule localization. The linkage error is a common problem in SMLM, and refers to the space between the fluorescent label and the target protein; thus, larger labels have a larger linkage error [57]. An early attempt relied on the incorporation of non-canonical amino acids (ncAAs) into a protein of interest by hijacking the normal stop-codon machinery in mammalian cells [58,59]. This allowed the incorporation of a variety of ncAAs, which in turn permitted the addition of azide functionalized target strands using click-chemistry to facilitate DNA-PAINT imaging. More recently, two approaches to achieve this have been published; Affimer labeling (Figure 5b) [60] and an aptamer-based approach (Figure 5d) [61]. Affimers are isolated from large phage-display libraries and are screened against the protein of interest for high specificity and affinity. This, coupled with their small size (~2 nm) makes them attractive for labeling biomolecules (Figure 5d) [60]. Thus far, DNA-PAINT imaging with these affimers has only been demonstrated for the actin cytoskeleton (Figure 5b). The other approach employed slow off-rate modified aptamers, so called SOMAmers (Figure 5d). Again the small size and high selectivity of the labels is attractive for super-resolution microscopy, as is the stoichiometric labeling of 1:1 target per label. SOMAmers were demonstrated for labeling of several cellular proteins, including EGFRs (Figure 5d). Here they were able to resolve EGFR receptor dimers in the plasma membrane (Figure 5d). In the same work a SOMAmer raised against GFP was used, thus, showing the potential flexibility for labeling within already existing modified cell lines [61].

### Current Challenges in DNA-PAINT

DNA-PAINT has proven to be a powerful tool for super-resolution microscopy. However, going forward, further development will be required to alleviate some technical challenges which face the method. First, it has been acknowledged in the field that one particular problem is non-specific binding, where imaging strands bind to off target sites, especially in biological settings such as fixed cells [38]. The recent development of FRET-PAINT tackled this head on and achieved background free imaging, given the requirement of both donor and acceptor to be present [38]. This has recently been extended additionally to use fluorescence lifetimes to discriminate between different docking species on an origami [50]. This approach has come at the cost of executing quantitative measurements, such as those discussed for qPAINT above, due in part to the additional complexity required to achieve FRET-based imaging.

Similarly, applying DNA-PAINT for quantitative measurements in cells has recently been of interest. Crucial for counting proteins using qPAINT is knowing the number of docking strands you have per protein (Figure 4a), which can become difficult depending on the conjugation method. For example, antibody labels will bear a distribution of docking strands given the conjugation method. Therefore, methods which can guarantee 1:1 stoichiometry of docking strand to molecules of interest are vital. This has been addressed recently, as discussed above, using ncAA incorporation, affimers, and SOMAmers which all allow 1:1 functionalization. However, although SOMAmers spend a long time bound, they still rely on a non-covalent interaction, and can potentially leave the target during long imaging times. Similarly, affimers are non-covalent, and sometimes require post-fixation, which may lead to off-target labeling. Finally, while ncAA incorporation does allow a covalent stoichiometric linkage, it suffers from low expression and efficiency for labeling. Therefore, new methods are required that combine the advantages of each of these labeling methods, i.e., covalent, small, highly specific, and stoichiometric. This would allow more challenging experiments that can observe and quantify proteins of low abundance in their native environment.

## 9. Conclusions

DNA-PAINT has greatly increased the use of DNA origami in the imaging community. In particular, it has opened the door for the use of DNA origami and DNA nanotechnology in cellular imaging and to resolve cellular interactions at the nanoscale. In return, the application of super-resolution imaging to DNA origami has offered detailed kinetic information about hybridization, incorporation, and stability of DNA origamis. As novel, dynamic applications appear, for example DNA-origami that undergoes a conformational change in response to a stimulus such as light, then DNA-PAINT and other imaging modalities will be central to verifying that the conformational changes occur as intended. Likewise, the demand for new and better probes and methods for cellular imaging will drive creative developments in the design of DNA origami.

## Figures and Tables

**Figure 1 genes-09-00621-f001:**
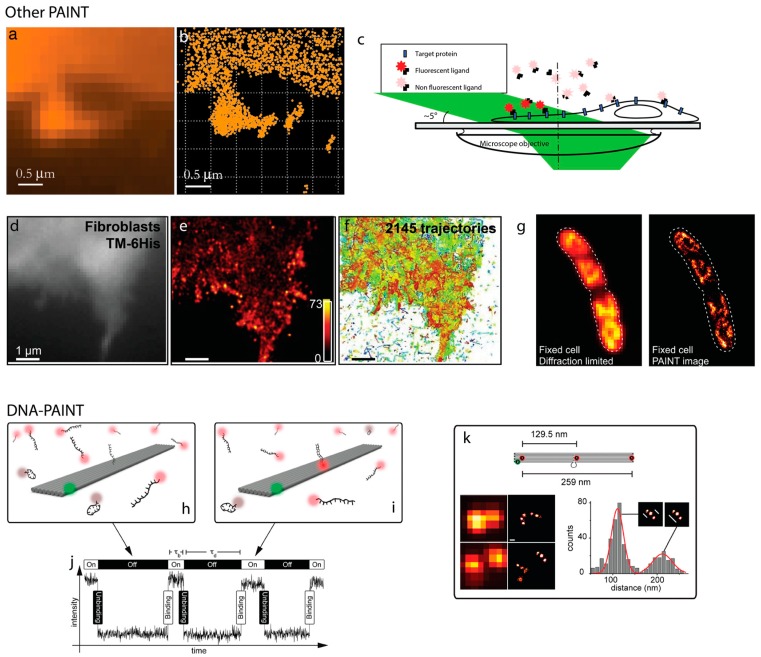
Examples of point-accumulated imaging for nanoscale topography (PAINT). (**a**) Initial implementation of PAINT from [10] showing raw fluorescence image from Nile Red imaging of a supported bilayer and (**b**) high-resolution image by the localization of 2778 single Nile Red probes collected in 4095 frames. (**c**) Schematic of universal PAINT (uPAINT) imaging showing exchange of sub-populations of fluorescent ligands, and super-resolution imaging (**d**) of TM-6His, transmembrane domain of the PDGF receptor fused to a six-histidine (6His) tag on its extracellular side, using 2045 trajectories (**e**). (**f**) Super-resolution PAINT imaging of green fluorescent protein (GFP)-Fis using JF646-Hoechst after 30 min FA fixation, adapted from [15]. (**g**) Schematic of DNA-PAINT. A DNA origami tile is marked with a green fluorophore and the docking strand (center of tile) is imaged when the imager-strand (red fluorophore) is transiently bound to the docking strand and, thus, the tile (**h**). Intensity versus time plots the time between binding events can be resolved (**i**). DNA-PAINT used to resolve the spatial separation of multiple docking sites (**j**). Images in (**b**–**e**) adapted from [11]; in (**g**–**j**) adapted from [17].

**Figure 2 genes-09-00621-f002:**
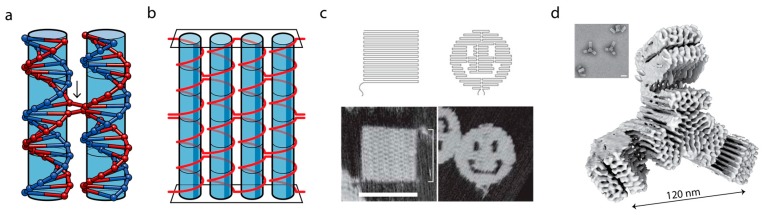
DNA Origami: (**a**) Basis of DNA origami is the Holliday junction, where a single strand of DNA crosses over to form a ‘crossover’ between adjacent helices. When two Holliday junctions sit along the same pair of helices this forms a double crossover. (**b**) If multiple Holliday junctions are arranged along multiple parallel helices this stitches the helices together to form a tile (adapted from [22]). (**c**) Original DNA origami tile designs and AFM from [23], showing tile and ‘three holed disc’. Scale bar = 100 nm. (**d**) CryoEM single particle reconstruction to 2.5 Å showing state of the art subunits for gigadalton assembly from [24]. Inset: representative field of view TEM micrograph, scale bar = 50 nm.

**Figure 3 genes-09-00621-f003:**
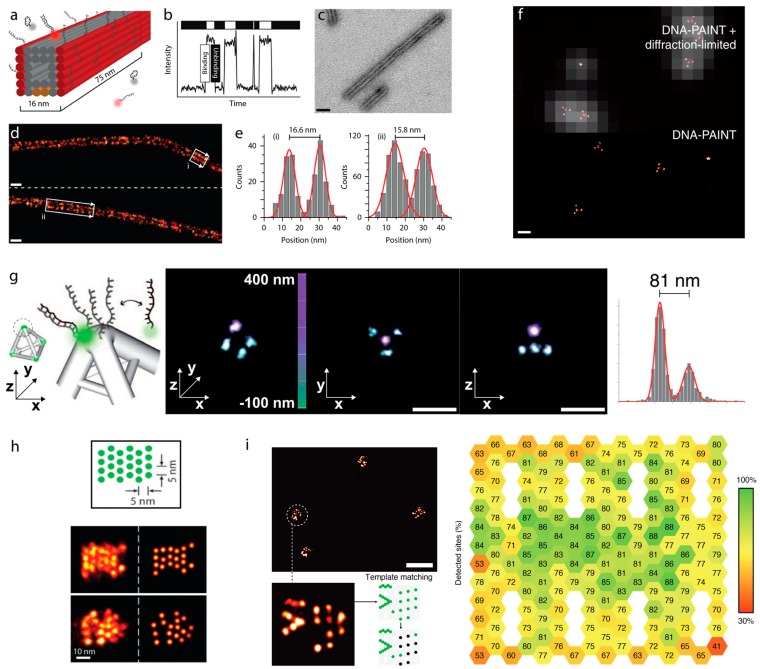
DNA-PAINT on DNA origamis. (**a**) A hollow interior 3D DNA origami polymer is decorated with single stranded docker strands on opposite faces. (**b**) Transient binding between imager and docker produces blinking that can be used for localization or kinetic quantification. (**c**) Transmission electron microscopy of the origami polymers (scale bar = 20 nm). (**d**) DNA-PAINT super-resolution image of the DNA origami polymer. (**e**) Histograms of cross-sections in boxed areas from (**d**). Images in (**a**–**e**) from [39]. (**f**) Overlay of diffraction limited and DNA-PAINT image of tetrahedron DNA-origamis (top), with the DNA-PAINT image alone showing spots corresponding to the vertices of the origami (scale bar = 200 nm). (**g**) 3D DNA-PAINT, achieved by SMLM imaging of PSF modified by a cylindrical lens of the same origamis in **f**) revealing that the length of each side of the tetrahedron could be accurately determined (scale bars = 200 nm). Images in (**f**,**g**) from [41]. (**h**) DNA-PAINT imaging to resolve sites spaced 5 nm apart on origami tile. Image from [44]. (**i**) DNA-PAINT imaging used to determine the incorporation efficiency of staples on an origami tile, with the percentage of staples detected in each position represented in the pictogram (left). Image from [45].

**Figure 4 genes-09-00621-f004:**
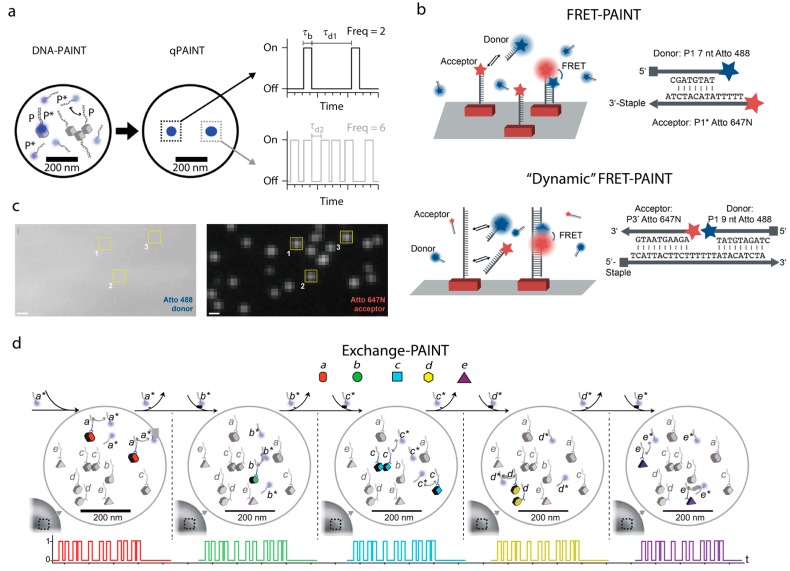
Extended applications of DNA-PAINT. (**a**) qPAINT. Two different complexes unresolvable by DNA-PAINT imaging with one or three docking strands. The two complexes, owing to the predictable binding of the imager strand to the docking strand, have characteristic blinking traces, with the complex possessing 3 docking strands having an increased frequency of binding events. By observing the time between events the number of docking strands can be determined from such traces given a single docking site calibration. Image adapted from [37]. (**b**) Fluorescence resonance energy transfer (FRET)-PAINT relies on bringing imager DNA with a donor fluorophore into proximity with DNA possessing an acceptor fluorophore. There are two routes for such imaging, one where the acceptor is fixed to the docking strand (top) and another where both the acceptor and donor are free in solution and require hybridization to a longer docking strand, termed ‘dynamic’ FRET-PAINT. (**c**) The FRET-PAINT approach allows very specific emission only when there is an acceptor present and in proximity short enough for FRET to be observed, which reduces the background from high imager concentration. As an example, the constant signal in the donor channel (left) can be compared with punctate signals in the acceptor channel when the donor is bound (right, boxes 1–3). Scale bars = 500 nm. Images adapted from [38]. (**d**) Exchange-PAINT relies on the ability to generate docking strand with mutually exclusive docking sequences (***a***–***e***). This means that each docking strand can only be sampled by its complementary imaging strand (***a****–***e****). Thus, repeated cycles of imaging, washing, and then imaging with a different complementary imager can be exploited to observe multiple species within a sample, and gives great scope for multiplexed imaging. Images adapted from [48].

**Figure 5 genes-09-00621-f005:**
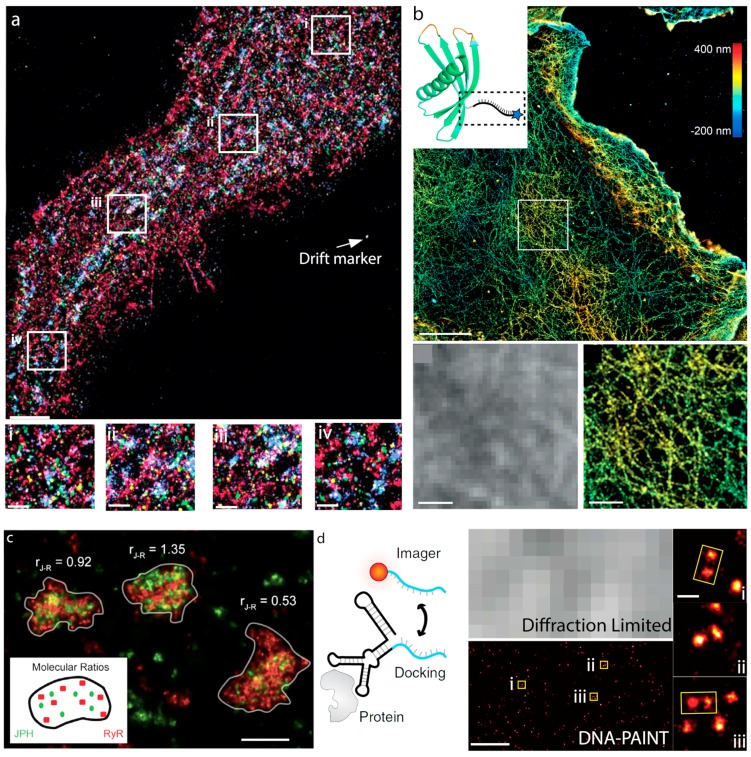
Cellular Imaging using DNA-PAINT. (**a**) Multiplexed Exchange-PAINT of EGFR (red), ErbB2 (green), ErbB3 (blue), IGF-1R (yellow) and Met (purple) in fixed BT-20 cells (top, scale bar-5 µm). Zoomed regions (**i**–**iv**) for different areas of the plasma membrane with the different protein imaged merged (scale bar = 1 µm). Images are from [48]. (**b**) 3D DNA-PAINT of the actin cytoskeleton using labeling with affimer probes (top; scale bar = 5 µm), inset—schematic of ribbon-structure of affimer probe with DNA docking strand attached. Comparison of diffraction limited image (left) from boxed region to 3D DNA-PAINT image (right). Scale bar = 1 µm. Images are from [60]. (**c**) qPAINT imaging of RyR (red) and junctophilin-1 (green) clusters. Numbers indicate ratios of junctophilin-1 to RyR (scale bar—250 nm). Images are from [47]. (**d**) SOMAmers for DNA-PAINT imaging of single proteins on fixed and live cells. Schematic of SOMAmers labeling and application to EGFRs in the plasma membrane (left). Comparison of diffraction limited image (top, middle) to the DNA-PAINT final convolved image (bottom, middle). Scale bar = 200 nm. EGFR dimers observed in the DNA-PAINT image, regions (**i**–**iii**) corresponding to identified dimers (scale bar = 20 nm). Images are from [61].

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
