# Peer review of "DNA-Based Super-Resolution Microscopy: DNA-PAINT"

_genes, 2018, doi:10.3390/genes9120621_

Round 1
Reviewer 1 Report
In their review "DNA-based super-resolution microscopy: DNA-PAINT", Nieves and co-workers discuss recent developments of super-resolution approaches based on the programmable, transient interaction of dye-labeled oligonucleotides to their target strands. Based on the initial PAINT implementation by Sharonov et al, the authors discuss recent advances in DNA-PAINT and related approaches from imaging synthetic DNA origami structures to cellular applications.
The review is comprehensive and generally well-written. I have a few remarks that the authors should address before the manuscript can be accepted for publication.
Line 16: Please make sure to use a consistent description for PAINT: "points accumulation for imaging in nanoscale topography"
Line 38: I would argue that this is true for localization precision, however not for localization accuracy. Precision depends mainly on the number of detectable photons (as stated). However for accuracy, the label size, orientation, and efficiency (for ultimate quality) needs to be considered as well. Please replace accuracy by precision
Line 41: I would recommend the authors to use STORM as acronym for stochastic reconstruction microscopy, initially introduced by the Zhuang lab (please cite the initial STORM paper from 2006: Rust, M. J., Bates, M., & Zhuang, X. (2006). Sub-diffraction-limit imaging by stochastic optical reconstruction microscopy (STORM). Nature Methods, 3(10), 793–796. http://doi.org/10.1038/nmeth929)), see also line 51
Line 56: Bleaching to a dark state is misleading. I would just say bleaching.
Figure 1: The panel labels are mixed up: c exists twice.
Line 128/129: 10^6 (Ms)^-1 is not a binding constant as suggested by the naming K_d, but a second-order association constant k_on. Please rephrase and correct the paragraph accordingly.
Line 130: I would suggest deleting the first sentence, as the rough description of FISH is not related to super-resolution microscopy.
Line 158: I would argue that characterization of dynamic DNA structures was not obtained in this example. I would rephrase to "... allowing characterization of the near-solution shape of DNA origami structures..."
Line 204: I would delete the part "...and when some of the docking strands are not incorporated in all origami...". qPAINT, or any related SMLM counting approach will not be able to recover the unlabeled fraction of molecules under any circumstances.
Line 258: Should read 10 unique docking strands instead of 9
Paragraph starting at Line 261: Please also cite https://doi.org/10.1186/s13041-017-0344-5.
Line 301: Should be HSP60 not HSP90
Line 375: Delete dynamics.
Author Response
In their review "DNA-based super-resolution microscopy: DNA-PAINT", Nieves and co-workers discuss recent developments of super-resolution approaches based on the programmable, transient interaction of dye-labeled oligonucleotides to their target strands. Based on the initial PAINT implementation by Sharonov et al, the authors discuss recent advances in DNA-PAINT and related approaches from imaging synthetic DNA origami structures to cellular applications.
The review is comprehensive and generally well-written. I have a few remarks that the authors should address before the manuscript can be accepted for publication.
We thank the reviewer for their kind comments and suggestions to improve our manuscript. We have addressed the individual comments below.
Line 16: Please make sure to use a consistent description for PAINT: "points accumulation for imaging in nanoscale topography"
- Now amended in abstract and throughout.
Line 38: I would argue that this is true for localization precision, however not for localization accuracy. Precision depends mainly on the number of detectable photons (as stated). However for accuracy, the label size, orientation, and efficiency (for ultimate quality) needs to be considered as well. Please replace accuracy by precision
- This has now been modified to remove discussion of ‘accuracy’ in favour of precision in text
Line 41: I would recommend the authors to use STORM as acronym for stochastic reconstruction microscopy, initially introduced by the Zhuang lab (please cite the initial STORM paper from 2006: Rust, M. J., Bates, M., & Zhuang, X. (2006). Sub-diffraction-limit imaging by stochastic optical reconstruction microscopy (STORM). Nature Methods, 3(10), 793–796. http://doi.org/10.1038/nmeth929)), see also line 51
- STORM is now the acronym used for stochastic reconstruction microscopy in text, and the original work has now been cited.
Line 56: Bleaching to a dark state is misleading. I would just say bleaching.
- ‘bleaching to dark state’ now removed in favour of ‘bleaching’.
Figure 1: The panel labels are mixed up: c exists twice.
- Now corrected.
Line 128/129: 10^6 (Ms)^-1 is not a binding constant as suggested by the naming K_d, but a second-order association constant k_on. Please rephrase and correct the paragraph accordingly.
- Now corrected.
Line 130: I would suggest deleting the first sentence, as the rough description of FISH is not related to super-resolution microscopy.
- First sentence now removed line 130.
Line 158: I would argue that characterization of dynamic DNA structures was not obtained in this example. I would rephrase to "... allowing characterization of the near-solution shape of DNA origami structures..."
- Now amended in accordance with above.
Line 204: I would delete the part "...and when some of the docking strands are not incorporated in all origami...". qPAINT, or any related SMLM counting approach will not be able to recover the unlabeled fraction of molecules under any circumstances.
- ‘and when some of the docking strands are not incorporated in all origami’ removed and replaced with reviewer’ suggestion.
Line 258: Should read 10 unique docking strands instead of 9
- Now corrected.
Paragraph starting at Line 261: Please also cite https://doi.org/10.1186/s13041-017-0344-5.
- Now cited.
Line 301: Should be HSP60 not HSP90
- Now corrected.
Line 375: Delete dynamics.
- Deleted.
Reviewer 2 Report
The manuscript by Nieves et al., 'DNA-based super-resolution microscopy: DNA-PAINT' reviews the history, development and recent advances of DNA-PAINT, which is a modality of super-resolution microscopy that uses the interaction between complementary DNA strands. DNA-PAINT is part of super-resolution microscopy technics relying on stochastic detections of individual fluorophores, Single Molecule Localization Microscopy (SMLM), as opposed to coordinate-targeted nanoscopy such as STED/RESOLFT.
The review starts with the invention of Point Accumulation for Imaging Nanoscale Topography (PAINT), originally used to image the plasma membrane, and its generalization with the development of universal PAINT (uPAINT), enabling to image any given biomolecule with a known binding partner (e.g. antibody, ligand, small-label …). The DNA-PAINT is a powerful extension of uPAINT using the interaction between complementary DNA strands. The authors explain in details the advantages, the strengths of DNA-PAINT in the context of DNA origami imaging and characterization, and in the context of cellular imaging. They also explain in detail variations of DNA-PAINT enabling quantitative Single Molecule Localization Microscopy (qPAINT), multiplexed multi-color imaging (exchange PAINT), and ways to decrease non-specific labeling in DNA-PAINT (FRET-PAINT).
This manuscript is well written and covers extensively the literature on DNA-PAINT. The main weakness of the manuscript is the discussion/explanation of DNA-PAINT limitations and potential artifacts that could be generated by DNA-PAINT imaging. In addition, certain concepts or conclusions should be explained in more details when possible.
Main concerns:
1. Of course, DNA-PAINT is a wonderful technic. But it will be interesting for the reader to be aware of the limitations and artifacts that are associated with DNA-PAINT, at least obvious ones. Some examples are listed below.
- Accessibility problem. The fact that PAINT, in general, uses extemporaneous labeling with an external fluorescent ligand could prevent efficient labeling of target proteins if incorporated in crowded macromolecular complexes.
- Undercounting of DNA-strands. The authors nicely explained that ‘DNA-PAINT imaging of origami structures has also revealed that the interaction of docking strand and imager could be altered depending on the location of the docking strand’. They should then explain how this could bias qPAINT experiments during cell imaging.
- Labeling of intracellular targets with DNA-PAINT requires cell fixation and permeabilization. Thus currently DNA-PAINT could not be used to image live cells, except if the target is a membrane protein bearing an extracellular docking strand.
- Limitation of DNA-PAINT (PAINT in general) when used to label endogenous proteins with antibodies. By using one or two layers of antibody (about 12 nm in size), some advantages of using DNA-PAINT are lost (qPAINT and nanometer resolution) since antibodies are multivalent and big compared to the spatial resolution that could be obtained with DNA-PAINT.
Note that some of these limitations are also valid for others SMLM technics (PALM, STORM).
2. In the abstract line 13 to 16: ‘The requirement to observe molecules multiple times during an acquisition has pushed the field to develop methods that allow the repeated, transient binding of a fluorophore to a target. This repetitive binding is then used to build an image via point accumulation for imaging nanoscale topography (PAINT).’
I would not completely agree with the idea that PAINT, in essence, enables multiple labeling of the same target. This is certainly true for DNA-PAINT, but not for PAINT and uPAINT. In PAINT, labeling of the membrane is random; in uPAINT labeling of the target could be one irreversible event. The authors should rather explain the principle of PAINT, which uses stochastic binding of a fluorescent ligand instead of stochastic photo-activation of a ‘permanent’ label, to obtain super-resolved images or diffusion maps. Then they could explain that using interaction between complementary DNA strands in the context of PAINT offers many advantages that they describe in this review.
3. Line 38: ‘so that each fluorophore is accurately localized over many imaging cycles, known as single molecule localization microscopy (SMLM)’
I am not sure that this will be valid for all SMLM modalities. For STROM it could be more advantageous to bleach rapidly the fluorophore. I would remove ‘over many imaging cycles’ that is mainly valid for DNA-PAINT.
4. Paragraph on PAINT: Would be nice to explain more the principle of PAINT: a pool of fast diffusing fluorescent ligands in the medium, that could be imaged only when bound to their targets on the cell, since this target is either immobile or slowly diffusing on the membrane. Could the authors also explain that in PAINT the labeling could be irreversible, this justifies the need for reversible labeling that is easily obtained by using DNA strands.
5. Line 72. Explain in more details that PAINT uses the stochastic labeling of the membrane. While universal PAINT (uPAINT) generalized this approach to specific biomolecules: using defined ligand/receptor pair: antibody-epitope (Giannone et al., Biophysical J 2010), receptor ligand (Winckler et al., Scientific Reports 2013), small tags (TrisNTA/6His (Giannone et al., Biophysical J 2010); Nanobody/GFP (Ries et al., Nat Methods 2012)). This enables to obtain super-resolved images in fixed cells, and diffusion maps of membrane proteins in living cells (Giannone et al, Biophysical J 2010; Nair et al., J Neuroscience 2013).
6. Line 306 to 308: I would not agree with this idea ‘DNA-PAINT imaging has also provided a chance to observe cell surface receptors, often difficult to observe with other SMLM (e.g., dSTORM or PALM) techniques due to labelling artefacts and low abundance.’ There are numerous studies using dSTORM and PALM to perform super-resolution on membrane proteins and receptors.
7. Line 316: The authors should explain and develop more on this idea ‘These approaches seek to reduce the linkage error in single molecule localisation (due to label size).’ This is a common problem of SMLM.
Minor concerns and suggestions:
- Line 33: circumvent rather than push past?
- Line 42: …photoactivatable and photoconvertible…
- Line 47: would be nice to cite articles from the Betzig’s lab that developed PALM.
- Line 66: Could the authors precise here that in the original PAINT study they used the Nile Red to label the membrane, in this specific case the membrane of LUVs.
- Line 79: mistake in ‘DNA provides is a natural’
- Line 88: what is TM-6His?
- Line 135: ‘More recently FISH has been used in combination with 3D structured illumination microscopy [31], an alternative implementation of super-resolution microscopy, where probes with single stranded DNA handles from between 30 – 1000 nt in length were used.’ This sentence is not completely clear. Would be good to define what is structured illumination. Here this sentence could be misleading since it seems that structured illumination is always based on imaging with DNA strands as permanent labels.
- Line 148: …useful metrology platform…
- Line 157: Could the authors explain why they used ‘dynamic origami structures’.
- Paragraph on qPAINT: could be nice to mention here as well the recent article of Jungmann (ref 53 Nat Methods) that demonstr ated the ability of DNA-PAINT to observe dimers of EGF receptors.
- Line 160 to 163: Could the authors explain exactly why DNA-PAINT performs better than dSTORM in term of spatial resolution.
- Line 278: could this sentence be rephrased ‘and it was found not only could the efficiency be used to differentiate docking sequences with different binding geometries, but also allowed super-resolution FRET imaging’.
- Line 291: add ref [45] again
- Line 297: How the 3D imaging was performed here? Explain why this could not be used to image thick samples as opposed to spinning disk confocal.
- Line 303: Could the authors explain how imaging was performed in ref [49] and explain the difference with confocal approaches.
- Line 375: Since DNA-PAINT could only image membrane proteins with extracellular labels, I would perhaps not emphasize on ‘resolve cellular dynamics’.
- Line 377: Could the authors specify which ‘dynamic applications for DNA origami’ they are referring to.
Author Response
We thank the reviewer for reading our review article with keen detail and for their useful suggestions to improve our manuscript which we have addressed below.
Main concerns:
1. Of course, DNA-PAINT is a wonderful technic. But it will be interesting for the reader to be aware of the limitations and artifacts that are associated with DNA-PAINT, at least obvious ones. Some examples are listed below.
- Accessibility problem. The fact that PAINT, in general, uses extemporaneous labeling with an external fluorescent ligand could prevent efficient labeling of target proteins if incorporated in crowded macromolecular complexes.
- In general, accessibility is only a problem for DNA-PAINT implemented with antibodies. However, this is a good point to raise, as it increasingly highlights the recent advances in DNA-PAINT labelling is covered in the “Current Challenges in DNA-PAINT section”. Therefore, a small amount of text has been added at the start of this section addressing the accessibility problem.
- Undercounting of DNA-strands. The authors nicely explained that ‘DNA-PAINT imaging of origami structures has also revealed that the interaction of docking strand and imager could be altered depending on the location of the docking strand’. They should then explain how this could bias qPAINT experiments during cell imaging.
- Whilst it has been observed that the position of the docking strand effected the binding kinetics, altered kinetics at this scale would not lead to a dramatic effect on the qPAINT analysis, since large variations are required for the qPAINT method to fail dramatically, i.e., not able to accurately discern integer strands, and imager concentration can be altered to accommodate expected ranges of stoichiometry. We have an experimental paper currently under consideration which demonstrates that qPAINT is robust to docker proximity as close as 3 nm. We will discuss with editor the possibility of citation as pre-print or amending to include citation. Currently, we have added text to indicate current tested limits of docker proximity (at least up to 6 nm). The reviewer here is concerned mostly with potential bias during cell imaging. The problems with cellular measurements are not caused by varying docker location on DNA origamis, but more by unknown proximity and stoichiometry of dockers when conjugated using antibodies or other methods. This caveat is currently mentioned currently in the text.
- Labeling of intracellular targets with DNA-PAINT requires cell fixation and permeabilization. Thus, currently DNA-PAINT could not be used to image live cells, except if the target is a membrane protein bearing an extracellular docking strand.
- This distinction is made clear now in the “Current Challenges” section.
- Limitation of DNA-PAINT (PAINT in general) when used to label endogenous proteins with antibodies. By using one or two layers of antibody (about 12 nm in size), some advantages of using DNA-PAINT are lost (qPAINT and nanometer resolution) since antibodies are multivalent and big compared to the spatial resolution that could be obtained with DNA-PAINT.
- This is covered in the very briefly in the “Current Challenges” relating to new smaller labelling methods and the potential ‘linkage error’ of larger labels. The text had been modified to make the point raised by the reviewer more clear.
2. In the abstract line 13 to 16: ‘The requirement to observe molecules multiple times during an acquisition has pushed the field to develop methods that allow the repeated, transient binding of a fluorophore to a target. This repetitive binding is then used to build an image via point accumulation for imaging nanoscale topography (PAINT).’
I would not completely agree with the idea that PAINT, in essence, enables multiple labeling of the same target. This is certainly true for DNA-PAINT, but not for PAINT and uPAINT. In PAINT, labeling of the membrane is random; in uPAINT labeling of the target could be one irreversible event. The authors should rather explain the principle of PAINT, which uses stochastic binding of a fluorescent ligand instead of stochastic photo-activation of a ‘permanent’ label, to obtain super-resolved images or diffusion maps. Then they could explain that using interaction between complementary DNA strands in the context of PAINT offers many advantages that they describe in this review.
- We thank the reviewer for the attention to clarity in our abstract. We have amended our abstract to make our intentions in this document more clear as follows:
“The requirement to observe molecules multiple times during an acquisition has pushed the field to explore methods that allow the binding of a fluorophore to a target. This binding is then used to build an image via points accumulation for imaging nanoscale topography (PAINT), which relies on the stochastic binding of a fluorescent ligand instead of the stochastic photo-activation of a permanently bound fluorophore. Recently, systems that use DNA to achieve repeated, transient binding for PAINT imaging have become the cutting edge in SMLM.”
3. Line 38: ‘so that each fluorophore is accurately localized over many imaging cycles, known as single molecule localization microscopy (SMLM)’
I am not sure that this will be valid for all SMLM modalities. For STROM it could be more advantageous to bleach rapidly the fluorophore. I would remove ‘over many imaging cycles’ that is mainly valid for DNA-PAINT.
– Now modified in accordance with reviewer suggestion.
4. Paragraph on PAINT: Would be nice to explain more the principle of PAINT: a pool of fast diffusing fluorescent ligands in the medium, that could be imaged only when bound to their targets on the cell, since this target is either immobile or slowly diffusing on the membrane. Could the authors also explain that in PAINT the labeling could be irreversible, this justifies the need for reversible labeling that is easily obtained by using DNA strands.
– reference to irreversible labelling now added.
5. Line 72. Explain in more details that PAINT uses the stochastic labeling of the membrane. While universal PAINT (uPAINT) generalized this approach to specific biomolecules: using defined ligand/receptor pair: antibody-epitope (Giannone et al., Biophysical J 2010), receptor ligand (Winckler et al., Scientific Reports 2013), small tags (TrisNTA/6His (Giannone et al., Biophysical J 2010); Nanobody/GFP (Ries et al., Nat Methods 2012)). This enables to obtain super-resolved images in fixed cells, and diffusion maps of membrane proteins in living cells (Giannone et al, Biophysical J 2010; Nair et al., J Neuroscience 2013).
– This discussion has been added, as have the reviewer’s suggestions for additional citations.
6. Line 306 to 308: I would not agree with this idea ‘DNA-PAINT imaging has also provided a chance to observe cell surface receptors, often difficult to observe with other SMLM (e.g., dSTORM or PALM) techniques due to labelling artefacts and low abundance.’ There are numerous studies using dSTORM and PALM to perform super-resolution on membrane proteins and receptors.
– This sentence had now been removed.
7. Line 316: The authors should explain and develop more on this idea ‘These approaches seek to reduce the linkage error in single molecule localisation (due to label size).’ This is a common problem of SMLM.
– Additional detail has been now added on this issue, and a new reference from a recent review on probes for SMLM. Moore, R.P.; Legant, W.R. Improving probes for super-resolution. Nat Methods 2018, 15, 659-660, doi:10.1038/s41592-018-0120-1.
Minor concerns and suggestions:
- Line 33: circumvent rather than push past?
- now changed in text.
- Line 42: …photoactivatable and photoconvertible…
- now changed in text.
- Line 47: would be nice to cite articles from the Betzig’s lab that developed PALM.
– original PALM paper now cited.
- Line 66: Could the authors precise here that in the original PAINT study they used the Nile Red to label the membrane, in this specific case the membrane of LUVs.
- This is stated in the next line(s); ‘The first iteration of this was by Sharonov & Hochstrasser [9], who used intermittent collisions between Nile Red and large unilamellar vesicles (LUVs) as the basis of their imaging (Fig. 1).’
- Line 79: mistake in ‘DNA provides is a natural’
- now changed in text.
- Line 88: what is TM-6His?
- “transmembrane domain of the PDGF receptor fused to a six-histidine (6His) tag on its extracellular side” now added to the figure legend for clarity.
- Line 135: ‘More recently FISH has been used in combination with 3D structured illumination microscopy [31], an alternative implementation of super-resolution microscopy, where probes with single stranded DNA handles from between 30 – 1000 nt in length were used.’ This sentence is not completely clear. Would be good to define what is structured illumination. Here this sentence could be misleading since it seems that structured illumination is always based on imaging with DNA strands as permanent labels.
- extra text added to provide clarity on 3D-SIM approach and combination with DNA-PAINT method.
- Line 148: …useful metrology platform…
- now changed in text.
- Line 157: Could the authors explain why they used ‘dynamic origami structures’.
- This part has been changed according to Reviewer 1’s suggestion.
- Paragraph on qPAINT: could be nice to mention here as well the recent article of Jungmann (ref 53 Nat Methods) that demonstr ated the ability of DNA-PAINT to observe dimers of EGF receptors.
- We thank the reviewer for this observation. This reference does indeed show the observation of EGFRs as dimers. However, this was not demonstrated by qPAINT, as per the 2006 paper, therefore, is not included in this section which focuses on qPAINT.
- Line 160 to 163: Could the authors explain exactly why DNA-PAINT performs better than dSTORM in term of spatial resolution.
- Extra text has now been added to improve the clarity of this.
- Line 278: could this sentence be rephrased ‘and it was found not only could the efficiency be used to differentiate docking sequences with different binding geometries, but also allowed super-resolution FRET imaging’.
- This has now been reworded and some detail added.
- Line 291: add ref [45] again
- This reference has been added.
- Line 297: How the 3D imaging was performed here? Explain why this could not be used to image thick samples as opposed to spinning disk confocal.
- Detail for the 3D imaging in the below reference has now been added in text, as has a reference for the deconvolution of the 3D data.
Beliveau, B.J.; Boettiger, A.N.; Avendano, M.S.; Jungmann, R.; McCole, R.B.; Joyce, E.F.; Kim-Kiselak, C.; Bantignies, F.; Fonseka, C.Y.; Erceg, J., et al. Single-molecule super-resolution imaging of chromosomes and in situ haplotype visualization using Oligopaint FISH probes. Nat Commun 2015, 6, 7147, doi:10.1038/ncomms8147.
- Line 303: Could the authors explain how imaging was performed in ref [49] and explain the difference with confocal approaches.
- Further details of ref 49 (now ref 56) has now been added. This should elucidate that this set-up is using a line-focused laser.
- Line 375: Since DNA-PAINT could only image membrane proteins with extracellular labels, I would perhaps not emphasize on ‘resolve cellular dynamics’.
- Now removed in text.
- Line 377: Could the authors specify which ‘dynamic applications for DNA origami’ they are referring to.
We have added an example to clarify this in the text: “As novel, dynamic applications appear, for example DNA-origami that undergoes a conformational change in response to a stimulus such as light, then DNA-PAINT and other imaging modalities will be central to verifying that the conformational changes occur as intended. Likewise, the demand for new and better probes and methods for cellular imaging will drive creative developments in the design of DNA origami.”
Reviewer 3 Report
In the review article “DNA-based super-resolution microscopy: DNA-PAINT” by Nieves et al. the authors give an up to date overview of the DNA-PAINT technology, including the development of the method as well as typical applications. In addition several modifications from the original approach are presented and discussed together with an outlook and further developments required. Taken together the article is well written, includes the current state of the literature and provides a comprehensive review of the current state both for experiences users as well as for people interested in getting started. The quality of the figures could be improved, possibly by getting original figures from the cited publications, rather then combining figures extracted from the publication files. The authors should also include some comments on DNA-PAINT applications in living cells and the problems associated with it. Some minor points are listed below.
Minor points:
Introduction: “ The accuracy of 38 localization depends only on the number of photons we can observe” insert “(N” after photons to correspond to the inline equation.
Introduction: “GFP has a low emission when excited at 488 nm and which increases…”, should it not read “ … 488 nm and increases …”
Introduction: “…can emit before bleaching to a dark state.” “…bleach to a permanently dark state” to highlight the difference to the blinking effect
PAINT: “on specific or tuneable interactions between” should read “tunable”
PAINT: “DNA provides is a natural candidate” should read “DNA provides a natural candidate”
Figure 1: The enumeration “c” is used twice in the figure
Sub-diffraction imaging of DNA origami via DNA-PAINT: “DNA origami structures by fluorescence-based microscopies, both” should read “microscopy”
Figure 3 caption: “3D DNA-PAINT imaging of the same origamis in d) revealing .. “ is there a mix-up with the sub-image reference? I suppose the origamis discussed here are shown in figure f? Additionally the authors should provide details on how the images in g were generated (3D).
Counting molecules/DNA-docking strands: “This ‘off-time’ can then be converted into as stoichiometry of…” “as” should read “a”
Counting molecules/DNA-docking strands: “…with a known number of sites[32].” Missing space before “[“
Figure 4 caption: “imager stand to the dicking strand” should read “docking”
Figure 4 caption: “…and in proximity long enough for FRET…” “long” should be replaced by “short”
Figure 4 caption: “The means the each can only be sampled….” This sentence requires re-writing.
Figure 4: in my version the image quality is very low, so details are lost.
Figure 5: the size of the sub-figure labels is variable. Sub-label “b and c” is present twice
Author Response
We thank the reviewer for their time and suggestions and have responded to their comments and suggestions below.
Minor points:
Introduction: “ The accuracy of 38 localization depends only on the number of photons we can observe” insert “(N” after photons to correspond to the inline equation.
- Now inserted in text.
Introduction: “GFP has a low emission when excited at 488 nm and which increases…”, should it not read “ … 488 nm and increases …”
- Now amended in text.
Introduction: “…can emit before bleaching to a dark state.” “…bleach to a permanently dark state” to highlight the difference to the blinking effect
- This has been amended in line with Reviewer 1’s suggestion.
PAINT: “on specific or tuneable interactions between” should read “tunable”
We have used UK English throughout for our language, and as such have left this as ‘tuneable’ here, but are happy to alter this if the journal has a preference for American English.
PAINT: “DNA provides is a natural candidate” should read “DNA provides a natural candidate”
- Now amended in text.
Figure 1: The enumeration “c” is used twice in the figure
- Now amended in figure.
Sub-diffraction imaging of DNA origami via DNA-PAINT: “DNA origami structures by fluorescence-based microscopies, both” should read “microscopy”
- Now amended in text.
Figure 3 caption: “3D DNA-PAINT imaging of the same origamis in d) revealing .. “ is there a mix-up with the sub-image reference? I suppose the origamis discussed here are shown in figure f? Additionally the authors should provide details on how the images in g were generated (3D).
- Details of 3D SMLM imaging using cylindrical lens now added to legend.
Counting molecules/DNA-docking strands: “This ‘off-time’ can then be converted into as stoichiometry of…” “as” should read “a”
- Now amended in text.
Counting molecules/DNA-docking strands: “…with a known number of sites[32].” Missing space before “[“
- Now amended in text.
Figure 4 caption: “imager stand to the dicking strand” should read “docking”
- Now amended in text.
Figure 4 caption: “…and in proximity long enough for FRET…” “long” should be replaced by “short”
- Now amended in text.
Figure 4 caption: “The means the each can only be sampled….” This sentence requires re-writing.
- Now amended in text.
Figure 4: in my version the image quality is very low, so details are lost.
- This should not be a problem in the final version, as all the final figures will be produced at high resolution.
Figure 5: the size of the sub-figure labels is variable. Sub-label “b and c” is present twice.
- This was amended in our original submission figures but may have been altered in the journal compilation process. We have now reamended in figure and will check with proofing and journal that the figures are represented correctly with correct sublabels in any final preprint edition.